# The High-Altitude Volcanic Caves of Mount Etna (Eastern Sicily): A Relevant Refuge for Some Ptero-Bryophyte Communities

Marta Puglisi [1], Giulia Bacilliere [1], Giulia Miraglia [1], Dario Teri [2] and Saverio Sciandrello [1,*]

[1] Department of Biological, Geological and Environmental Sciences, University of Catania, Via A. Longo 19, I-95125 Catania, Italy; mpuglisi@unict.it (M.P.); giulia.bacilliere@phd.unict.it (G.B.); giuliam276@gmail.com (G.M.)

[2] Via Vincenzo Bellini, 35 B, Tremestieri Etneo, I-95030 Catania, Italy; dario@etnasci.it

* Correspondence: s.sciandrello@unict.it

**Abstract:** A phytosociological analysis of the plant communities in high-altitude volcanic caves of Mount Etna (Sicily), based on literature data and unpublished relevés, is presented. A total of 147 phytosociological relevés were processed and analyzed using classification and ordination methods. Classification of the relevés, supported by ordination, showed two main vegetation groups: the first one includes communities of the *Pohlion crudae* alliance, and the other includes the vegetation from the *Pohlio crudae–Asplenion septentrionalis* alliance. Furthermore, two new communities, *Pohlio crudae–Cystopteridetum dickieanae* subass. *amphidietosum mougeotii* and *Pohlio crudae–Cystopteridetum dickieanae* subass. *polystichetosum lonchitis*, are proposed for Etna. The high-altitude caves can be considered a refuge for these ptero-bryophytic rare communities.

**Keywords:** mountain habitat; phytosociology; plant communities; plant conservation; syntaxonomy; vegetation

## 1. Introduction

Caves represent a peculiar habitat of great geological and biological interest. For most taxonomic groups, caves represent one of the most challenging ecosystems due to their extreme conditions. According to the Natura 2000 network of the European Union (Habitats Directive 92/43/EC), caves fall within the Habitat 8320 (fields of lava and natural excavations) and the Habitat 8310 (caves not open to the public). Caves are characterized by peculiar environmental factors, which determine the selection of flora and vegetation. Among these plants, bryophytes and ferns represent the dominant taxonomic group in these habitats [1–3], capable of adapting to the difficult ecological conditions of the cave habitat [4]. Due to the consistently stable humidity and temperature of the cave habitat and the low point of compensation for the light, some species can adapt to the lack of light with marked scio-morphoses and crypto-morphoses, as detected in species living in the poorly light areas of caves [3,4]. Ferns are less tolerant, but some of them, nevertheless, can live in this type of habitat together with bryophytes.

Mt. Etna, located in eastern Sicily, is a geologically recent volcano whose formation dates back to the Late Quaternary, having formed 500 ka ago along a diverging zone in the framework of the Africa–Europe plate convergence [5,6], with eruptions occurring beneath the sea off the ancient coastline of Sicily. It is very interesting for studying plant colonization processes, which are favored by its important altitudinal development (highest peak at 3357 m a.s.l.), geographic isolation, geo-lithological isolation, and the incessant volcanic activity leading to a continuous creation of new, bare land. The volcanic activity in the area led to the formation of different lava fields that hide caves formed by lava flows. As a result of the basic composition of magma and the mainly effusive volcanic activity, most of the formations are lava tubes. Mt. Etna hosts a peculiar flora with noteworthy

species of high phytogeographical interest, making this volcano an important biodiversity hot spot in the Mediterranean region [7–9].

The aim of this paper is to provide updated knowledge on the plant communities of the high-altitude volcanic caves of Mount Etna, allowing the findings of a remarkable set of bryophytes and pteridophytes.

## 2. Materials and Methods

### 2.1. Study Area

Mount Etna is a large volcano of basaltic nature with a surface of 1178 Km$^2$ and an altitude of 3357 m a.s.l. It is the highest active volcano in Europe and represents the highest mountain in Sicily. Due to its outstanding biodiversity and geological features, it was declared a Natural Heritage of Humanity by UNESCO in 2013. On Mt. Etna, over 200 lava caves are known [10], most of which are lava tubes, while the others are eruptive fractures, and very few of them are created by erosion [11,12]. The lava caves can be classified into pneumatogenetic or rheogenetic caves: the former originated by explosions, violent or gradual, of magmatic gas or superheated water vapor; the latter were generated by lava flowing on the surface (lava tubes). To explain the origin of the lava tubes, some researchers believe that once the volcanic activity is over, the still-fluid lava contained inside the tube flows downhill by gravity, causing the formation of lava-flowing caves. Other researchers hypothesize that as lava flows inside its conduit, it erodes the channel bottom with a progressive lowering of the floor of the lava tube. Finally, the simultaneous formation of a vacuum above the flow constitutes the tunnel. Although it is believed that lava tubes only form on pahoehoe lavas, caves on Mt. Etna frequently occur in "aa" lava, and some of the tubes are considered to have formed on "aa" lava flows which were subsequently covered by late-stage pahoehoe lava [13]. The composition of these caves corresponds mainly to basaltic lava formed by silicates such as clinopyroxene, plagioclase, and olivine, in addition to iron minerals intercalated with carbonates and opals of biogenic origin [12]. The lava caves and small lava tunnels reported in this paper are on Mt. Etna at altitudes ranging from 1350 m to 2200 m a.s.l.

### 2.2. Phytosociological Data and Statistical Analysis

The field surveys were carried out mostly from spring 2021 to autumn 2023. A total of 147 phytosociological relevés x 83 species were collected, of which 125 were from the literature [14–16] and 22 were unpublished. According to Tosco [17,18], Cortini Pedrotti [19], and Puglisi et al. [17], we have distinguished an outside area corresponding to the entrance (E), a liminar zone (L) from the entrance to a depth where the light reaches ½ of the external light intensity, and a subliminar zone (SL), with even more reduced brightness (less than ½ of the external light intensity). The original Braun-Blanquet sampling scale was transformed into the ordinal scale, according to Van der Maarel [20]. All the relevés were analyzed using classification methods. A multivariate analysis (Linkage method: Ward's, Distance measure: Euclidean) was applied. Cluster analyses were performed using PC-ORD 6 software Version 6 [21].

### 2.3. Plant Communities Processed

The processed relevés from the literature belong to the following syntaxa: *Pohlietum crudae* subass. *timmietosum bavaricae, Pohlio annotinae–Brachythecietum velutini, Pohlio annotinae–Brachythecietum velutini, Bartramietum ithyphyllae, Pohlio crudae–Amphidietum mougeotii, Pohlietum crudeae* [14,16]; *Pohlio crudae–Cystopteridetum dickieanae, Brachyteco velutini–Asplenietum trichomanis, Asplenio septentrionalis–Dryopteridetum villari* [15].

### 2.4. Taxonomic and Syntaxonomic Nomenclature

Vascular species nomenclature and identification follows the second edition of the Flora d'Italia [22]; species nomenclature of bryophytes follows Aleffi et al. [23].

Syntaxonomic classification follows Puglisi and Privitera [24] and Mucina et al. [25]. The names of syntaxa comply with the International Code of Phytosociological Nomenclature (ICPN) [26].

## 3. Results

### 3.1. Cluster Analysis

A total of 10 plant communities belonging to two phytosociological classes were identified by cluster analysis (Figure 1, Appendix A). Classification of the relevés, supported by cluster analysis, showed two main vegetation groups (Figure 1): cluster A corresponding to the alliance *Pohlion crudae*, which includes the sub-hygrophilous and sciophytic bryophyte communities on volcanic substrate in the supra-Mediterranean and oro-Mediterranean belts of Sicily; cluster B corresponding to the alliance *Pohlio crudae–Asplenion septentrionalis*, which includes the fern-rich chasmophytic vegetation of siliceous rock crevices in the supra-Mediterranean belt of Sicily and Calabria region (southern Italy). Within the *Pohlion crudae* alliance, four subgroups (with six communities) can be distinguished: the first one includes the *Pohlia cruda*-dominated communities (*Pohlietum crudae* subass. *timmietosum bavaricae* and *Pohlietum crudae* subass. *typicum*); the second group, the *Pohlio crudae–Amphidietum mougeotii*; the third group, the *Bartramietum ithyphyllae* community; and the last group, the *Brachytheciastrum velutinum*-dominated communities (*Pohlio annotinae–Brachythecietum velutini* and *Brachytecio velutini–Asplenietum trichomanis*). Within the *Pohlio crudae–Asplenion septentrionalis* alliance, four plant communities can be distinguished: the first two communities (*Pohlio crudae–Cystopteridetum dickieanae* subass. *polystichetosum lonchitis* and *Pohlio crudae–Cystopteridetum dickieanae* subass. *amphidietosum mougeotii)* are linked to the highest altitudes and survive in harsh environmental conditions, while *Pohlio crudae–Cystopteridetum dickieanae* subass. *typicum* and *Asplenio septentrionalis–Dryopteridetum felix-max* prefer less severe environmental conditions.

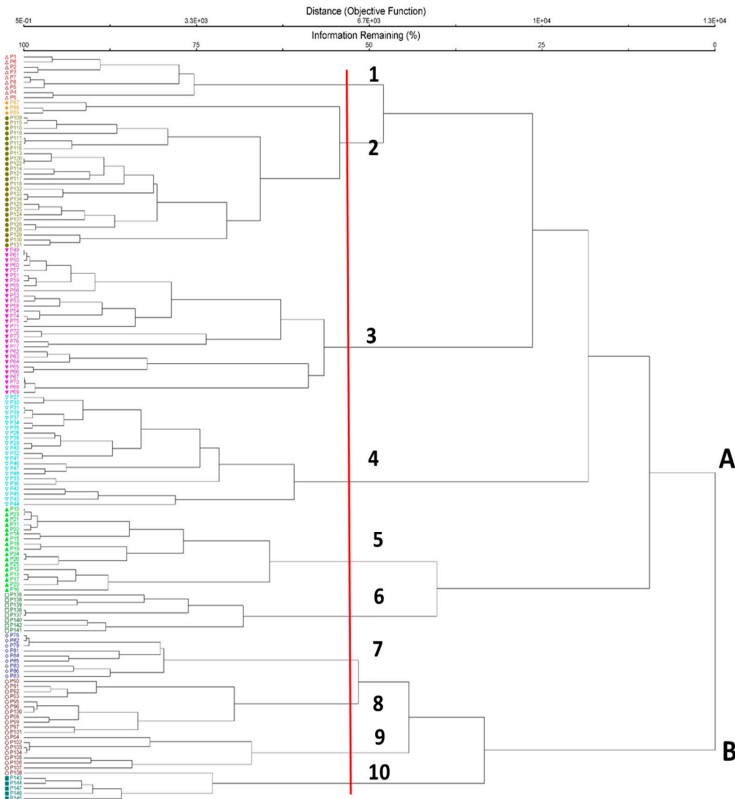

**Figure 1.** Cluster analysis of cave vegetation of the high-altitude Mount Etna. Plant communities: 1. *Pohlietum crudae* subass. *timmietosum bavaricae*; 2. *Pohlietum crudae* subass. *typicum*; 3. *Pohlio crudae–Amphidietum mougeotii*; 4. *Bartramietum ithyphyllae*, 5. *Pohlio annotinae–Brachythecietum velutini*;

6. *Brachytecio velutini–Asplenietum trichomanis*; 7. *Pohlio crudae–Cystopteridetum dickieanae* subass. *polysticheto-sum lonchitis*; 8. *Pohlio crudae–Cystopteridetum dickieanae* subass. *amphidietosum mougeotii*; 9. *Pohlio crudae–Cystopteridetum dickieanae* subass. *typicum*; 10. *Asplenio septentrionalis–Dryopteridetum filix-max.*

*3.2. Plant Community Description*

The floristic composition (Appendix B), ecology, syndynamic relationships, and chorology of each examined plant community are critically described below.

Alliance *Pohlion crudae* Privitera and Puglisi

Holotypus: *Pohlietum crudae* Privitera and Puglisi 1996

Diagnostic species: *Isopterygiopsis pulchella*, *Pohlia elongata*, *P. cruda*, *Bartramia ithyphylla*, *Brachytheciastrum collinum*, *B. velutinum*, *Amphidium mougeotii*.

Structure and ecology: This alliance group is montane and high-montane, sciophytic to markedly sciophytic, mesophytic or meso-hygrophytic bryophyte communities colonizing acid substrate within volcanic caves or in undergrowth slopes, niches, cracks, and fissures of rocks with accumulated soil.

Distribution: Mt. Etna at 1200–2350 m of altitude [14]; Southern Italy, Greece (unpublished data).

Notes: Mucina et al. [25] considered the alliance *Pohlion crudae* as a synonym of the *Pogonation urnigeri* v. Krusenstjerna 1945. Krusenstjerna [27] reports for Sweden a "*Pogonatum–Atrichum union* (*Pogonation*)", with floristic composition and ecology different from those of *Pohlion crudae*. He reports this "*Pogonation union*" colonizing clay-mixed sand or fine gravel in gravel pits, ditches, and road cuts. The floristic set is very heterogeneous with species from dry habitats and species from humid habitats, with a dominance of *Dicranella crispa*, *Ceratodon purpureus*, and *Polytrichum juniperinum*; these species are diagnostic for two different bryosociological classes, *Ceratodonto–Polytrichetea piliferi* Mohan 1978 and *Cladonio digitatae–Lepidozietea reptantis*. Therefore, we agree with Marstaller [28,29], who considered the syntaxon of Krusenstjerna as the suballiance *Pogonatenion urnigeri* (v. Krusenstjerna 1945) Philippi 1956 and as a synonym of the alliance *Dicranellion heteromallae* Philippi 1963.

3.2.1. *Pohlietum crudae* Privitera and Puglisi 1996 subass. *timmietosum bavaricae* Privitera and Puglisi 1996

Holotypus: Rel. 2, Table 11a [14]

Diagnostic species: *Timmia bavarica*

Structure and ecology: It is a bryocoenosis with dominance of acrocarpous mosses with caespitose habitus. In this community, the dominant life form is tall turf, and the prevalent life strategy is perennial stayer with high reproductive effort [16]. Perennial stayers show a long life span and a rather low sexual and asexual reproductive effort with a variable age of first reproduction—but several years at least. The spores are small (<25 μm in diameter), providing better chances of dispersal [30]; in general, the gametophytes are stress tolerant, competitive, and show a high morphological plasticity [30–32]. As concerns the ecology, *Pohlietum crudae subass. timmietosum bavaricae* behaves as a terricolous, meso-hygrophytic, sciophytic community.

Distribution: Lamponi cave (1750 m a.s.l.), one of the longest lava tunnels on Mt. Etna (400 m long). In this cave, in conditions of greater edaphic humidity, this community is replaced by *Bartramietum ithyphyllae*, and, in conditions of greater shade, by *Pohlio crudae-Ampidietum mougeotii*.

Notes: *Timmia bavarica* is a subcontinental-dealpine species occurring in Sicily only on Mt. Etna and Mt. Madonie; on Mt. Etna, it is found only within the Lamponi cave.

### 3.2.2. *Pohlietum crudae* Privitera and Puglisi 1996 subass. *typicum*

Holotypus: Rel. 17, Table 11 [15]

Diagnostic species: *Pohlia cruda* (Hedw.) Lindb.

Structure and ecology: It is a moss community with acrocarpous moss dominance, caespitose habitus, and a turf-life form. The prevalent life strategy is represented by colonist species. This life strategy is characterized by a generally low longevity (often pauciennial), an often high asexual reproduction by rhizoid gemmae and leaf gemmae for a rapid establishment, a regular formation of sporophytes, and the production of numerous small spores (<25 μm in diameter) [30–32]. Ecologically, *Pohlietum crudae* subass. *pohlietosum crudae* behaves as a terricolous, meso-hygrophytic, photo-sciophytic community.

Distribution: A few meters after the entrance of Lago cave (2200 m a.s.l.), Grotta degli Inglesi (1690 m a.s.l.), and in rocky cracks and small lava tunnels at Rifugio Citelli, Mt. Baracca, Contrada Germaniera, i Dammusi, and Mt. Palestra (1500–1950 m a.s.l.).

Notes: The community is characterized by *Pohlia cruda*, a boreo-arctic montane species, reaching, on Mt. Etna, the record altitude of 2500 m a.s.l. Within the floristic composition of this community, is emphasized the occurrence of *Grimmia torquata*, another interesting boreo-arctic montane species rare in Italy, occurring in Sicily only in some caves of Mt. Etna.

### 3.2.3. *Pohlio crudae–Amphidietum mougeotii* Privitera and Puglisi 1996

Holotypus: Rel. 8, Table 13 [15]

Diagnostic species: *Amphidium mougeotii*.

Structure and ecology: It is a terricolous, meso-hygrophytic, markedly sciophytic community. This troglophilous community characterizes the innermost part of Etna's high-altitude caves. The dominant life form is short turf; the prevalent life strategy is a perennial stayer with moderate reproductive effort and production of small spores [16]. In particular, sexual reproduction of species is quite rare, while asexual reproduction is favored (e.g., *Amphidium mougeotii*, *Isopterygiopsis pulchella*). In the less shaded parts of the Lamponi cave, it is replaced by *Pohlietum crudae* subass. *timmietosum bavaricae*.

Distribution: Lamponi, Palombe, Ladri, Casa del Vescovo, Cassone, Coniglio, Faggi and Intraleo caves, and small lava tunnels at Mt. Calcarazzi (1350–2200 m a.s.l.).

Notes: This community is dominated by *Amphidium mougeotii*, a boreal-montane species, rare in central and southern Italy, occurring in Sicily only within the caves of Mt. Etna. Among the species characterizing this association, it is to emphasize the occurrence of *Isopterygiopsis pulchella*, a strongly sciophytic species occurring in Sicily only within some caves of Mt. Etna, and *Brachytheciastrum collinum*, an Arctic-montane species very rare in Italy, found in Sicily only on Mt. Etna [8]. The last species is very rare in Europe as well as in the Mediterranean region [33,34].

### 3.2.4. *Bartramietum ithyphyllae* v. Krusenstjerna 1945

Lectotypus: Rel. 5, Table 38 [29]

Diagnostic species: *Bartramia ithyphylla*

Structure and ecology: It is a chasmophytic, acidophytic, mesophytic, sciophytic, and photo-sciophytic association, occurring at the entrance and a few meters after the cave entrance. This community is almost exclusively composed of acrocarpous mosses with short turf-life forms. The prevalent life strategy is the perennial shuttle due to the dominance of the characteristic species *Bartramia ithyphylla*; larger spores (> 25 μm in diameter up to 40 μm), decreasing long-range dispersal, and gametophyte longevity characterize this life strategy [16]. In conditions of lower edaphic humidity, it is replaced by *Pohlietum crudae* subass. *timmietosum bavaricae*.

Distribution: Ladri, Lamponi, and Gelo caves at altitudes of 1540–2046 m a.s.l.

Notes: *Bartramia ithyphylla* is a boreo-arctic montane species reaching an altitudinal record of 2500 m on Mt. Etna. This species is accompanied by a set of species of phytogeographical interest, such as the above-mentioned *Isopterygiopsis pulchella* and *Brachythecias-*

*trum collinum*, besides *Mielichhoferia mielichhoferiana*, a boreal-montane species occurring in Sicily only on Mt. Etna [8].

3.2.5. *Pohlio annotinae–Brachythecietum velutini* Privitera and Puglisi 1996

Holotypus: Rel. 2, Table 14 [14]

Diagnostic species: *Brachytheciastrum velutinum*

Structure and ecology: It is a xero-mesophytic, photo-sciophytic to sciophytic association. This bryocoenosis is characterized by *Brachytheciastrum velutinum*, a temperate moss species. The dominant life form is weft due to the dominance of the pleurocarpous moss component; the life strategy is perennial stayers with high sexual reproductive effort. In the subliminar zone of Ladri cave, it is replaced by *Pohlio crudae–Amphidietum mougeotii*.

Distribution: Tre Livelli and Ladri caves (1620–1540 m a.s.l.).

3.2.6. *Brachytecio velutini–Asplenietum trichomanis* Brullo and Siracusa in Brullo et al. 2004 [15]

Holotypus: rel. 4, Table 23 [15].

Diagnostic species: *Brachytheciastrum velutinum*

Structure and ecology: This chomophilous community is dominated by bryophytes and pteridophytes in the faintly shaded underwood rocky outcrops. It is characterized by some temperate and southern-temperate moss species, such as *Brachytheciastrum velutinum*, *Tortula subulata*, and *Homalothecium sericeum*, while the pteridophytic component is mainly represented by *Asplenium trichomanes*, *A. septentrionale*, and *A. ceterach*. The *Brachytecio velutini–Asplenietum trichomanis*, above 1700 m a.s.l. is replaced by the *Pohlio crudae–Cystopteridetum dickieanae*.

Distribution: Monte Intraleo, Tarderia, between 1000 and 1500 m a.s.l. [15].

Notes: This community is very similar and could probably be considered synonymous with *Pohlio annotinae–Brachythecietum velutini*, with which it shares the same ecology, altitudinal range, and floristic composition; the only difference is the presence of some ferns. These data are supported by the results of the cluster analysis.

Alliance *Pohlio crudae–Asplenion septentrionalis* Brullo and Siracusa in Brullo et al. 2004 [15].

Holotypus: *Pohlio crudae–Cystopteridetum dichkeianae* Brullo and Siracusa in Brullo et al. 2004 [15].

Diagnostic species: *Amphidium mougeotii*, *Bartramia pomiformis*, *Brachyteciastrum velutinum*, *Pohlia cruda*, *P. annotina*, *Tortula subulata*, *Asplenium septentrionale*.

Structure and ecology: Orophilous and siliceous vegetation dominated by bryophytes and pteridophytes growing in caves or shaded rocks. The *Pohlio crudae–Asplenion septentrionalis* groups the bryo-pteridophytic communities of shaded rocky surfaces in the high mountain belt between 1400 and 2200 m a.s.l. At low altitudes (0–1000 m a.s.l.), the *Pohlio crudae–Asplenion septentrionalis* alliance is replaced by *Bartramio–Polypodion serrati* [15].

Distribution: Etna and Aspromonte massif [15,35].

3.2.7. *Pohlio crudae–Cystopteridetum dickieanae polystichetosum lonchitis* subass. nova hoc loco

Holotypus: Rel. 7, Table 1, hoc loco.

Diagnostic species: *Polystichum lonchitis*.

Structure and ecology: This vegetation, exclusive to Sicily, colonizes the entrance of a small volcanic cave, reaching the innermost part of it, where it grows on very inclined and glassy rock walls. The cave, formed after the eruption of 1602, is placed at 1879 m a.s.l. on the western side of Mt. Etna. This bryo-pteridophyte community is dominated by the ferns *Cystopteris dickieana* and *Polystichum lonchitis* and by some mosses, such as *Pohlia cruda*, *Polytrichastrum alpinum*, *Sciuro-hypnum starkei*, *Brachytheciastrum collinum*, and *Bartramia ithyphylla*. This community prefers shade and cold environmental conditions, such as prolonged periods of snow. This community shows a troglophilous character,

representing the only bryo-pteridophyte community capable of reaching the deepest zones of high-altitude volcanic caves.

**Table 1.** *Pohlio crudae–Cystopteridetum dickieanae polystichetosum lonchitis* subass. nova hoc loco. According to the Braun-Blanquet method in each relevé, the complete list of plant species was recorded, and for each species, the cover value (percentage of soil surface) was assessed (+: <1% cover; 1: 1–5% cover; 2: 5–25% cover; 3: 25–50% cover; 4: 50–75% cover; 5: >75% cover). Asterisk (*) indicates the holotypus of the new association. Dates and locations of relevés: Rel. 1–4, Dagalotti del Diavolo (Etna W), 21.06.22, Sciandrello S.; Rel.5, Dagalotti del Diavolo (Etna W), 15.09.23, Sciandrello S.; Rel. 6–9, Dagalotti del Diavolo (Etna W), 20.10.23, Sciandrello and Bacilliere.

| Rel. numbers | 1 | 2 | 3 | 4 | 5 | 6 | 7 * | 8 | 9 | |
|---|---|---|---|---|---|---|---|---|---|---|
| Rel. number in dendrogram | 78 | 79 | 80 | 81 | 82 | 83 | 84 | 85 | 86 | |
| Area (m$^2$) | 1 | 1 | 1 | 1 | 1 | 1 | 1 | 1 | 1 | |
| Cover total (%) | 85 | 80 | 80 | 85 | 85 | 85 | 90 | 85 | 85 | |
| Aspect | SW | W | W | W | W | S | N | W | W | |
| Slope (°) | 80 | 80 | 80 | 10 | 85 | 80 | 90 | 15 | 85 | |
| Altitude (m s.l.m.) | 1879 | 1879 | 1879 | 1879 | 1879 | 1877 | 1877 | 1877 | 1877 | |
| Species number | 4 | 5 | 5 | 7 | 5 | 8 | 7 | 7 | 10 | |
| Volcanic flow age | 1602 | 1602 | 1602 | 1602 | 1602 | 1602 | 1602 | 1602 | 1602 | Freq. |
| **Differential species** | | | | | | | | | | |
| *Polystichum lonchitis* (L.) Roth | 2 | 1 | 2 | 4 | 1 | 2 | 2 | 3 | 1 | 9 |
| **Char. Pohlion crudae** | | | | | | | | | | |
| *Pohlia cruda* (Hedw.) Lindb. | 2 | 2 | 1 | 1 | 1 | 2 | 2 | 1 | 2 | 9 |
| *Brachytheciastrum collinum* (Schleich. ex Müll.Hal.) Ignatov and Hutten | + | 1 | 1 | 2 | 1 | 2 | 2 | 2 | 1 | 9 |
| *Bartramia ithyphylla* Brid. | . | 1 | . | 1 | . | . | + | 1 | 1 | 5 |
| *Pohlia elongata* Hedw. | 1 | 1 | . | . | 1 | 2 | . | 1 | . | 5 |
| *Isopterygiopsis pulchella* (Hedw.) Schimp. | . | . | . | . | . | 1 | . | . | 1 | 2 |
| **Char. Pohlio crudae–Asplenion septentrionalis** | | | | | | | | | | |
| *Cystopteris dickieana* R.Sim | 4 | 4 | 4 | + | 4 | 3 | 4 | 2 | 3 | 9 |
| **Other species** | | | | | | | | | | |
| *Sciuro-hypnum starkei* (Brid.) Ignatov and Hutten | . | . | 1 | 1 | . | 1 | . | + | 1 | 5 |
| *Polytrichastrum alpinum* (Hedw.) G.L.Sm. | . | . | 2 | 1 | + | 3 | . | . | 1 | 5 |
| *Hymenoloma crispulum* (Hedw.) Ochyra | . | . | . | 1 | . | . | 1 | . | . | 2 |
| *Rumex scutatus* L. subsp. *aetnensis* (C.Presl) Cif. and Giacom. | . | . | . | . | . | . | . | . | + | 1 |
| *Senecio aethnensis* Jan ex DC. | . | . | . | . | . | . | + | . | . | 1 |
| *Tanacetum vulgare* L. subsp. *siculum* (Guss.) Raimondo and Spadaro | . | . | . | . | . | . | . | . | + | 1 |

Distribution: Lava tube of Dagalotti del Diavolo (W Etna).

Notes: *Polystichum lonchitis* (Figure 2) is a rhizomatous geophyte with a circumboreal distribution, widespread in all regions of Italy except Puglia, Sardinia, and Sicily. The species was, for the first time, detected on Mt. Etna (Sicily) by one of the authors (Teri D.) in 2022 during an excursion. This very rare fern is considered a cold climate relict species, which probably arrived in Sicily during the last glaciations, finding suitable conditions for its survival inside a volcanic cave on Mt. Etna.

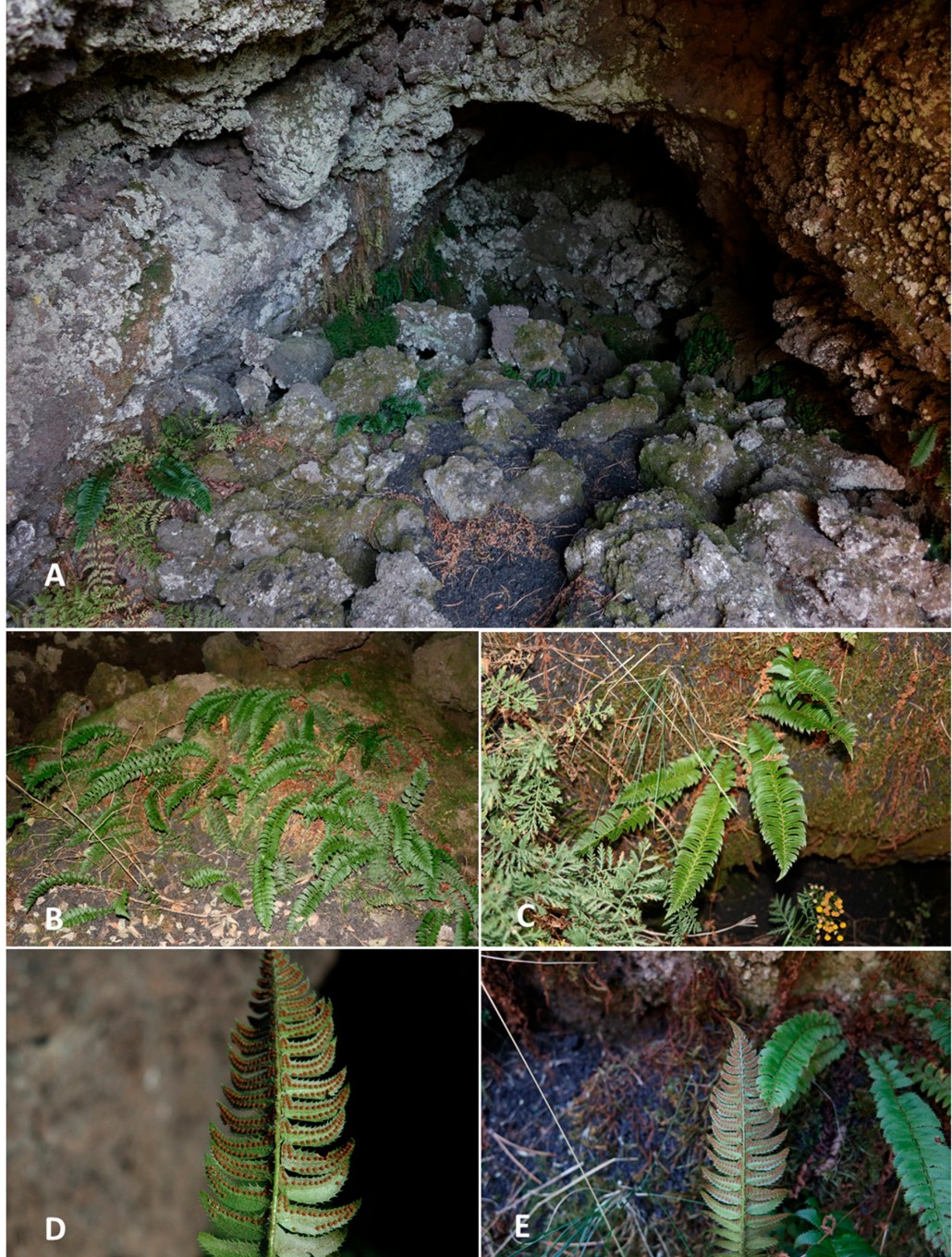

**Figure 2.** Photo plate illustration of *Polystichum lonchitis* in its natural habitat (Mt. Etna). (**A**) "Dagalotti del Diavolo" Cave; (**B**) *Polystichum lonchitis* at the base of the cave; (**C**) *Polystichum lonchitis* on the walls of the cave; (**D**,**E**) detail of sporangia.

3.2.8. *Pohlio crudae–Cystopteridetum dickieanae* Brullo and Siracusa in Brullo et al. 2004 [15] *amphidetosum mougeotii* subass. nova hoc loco

Synonyms: *Pohlio crudae–Cystopteridetum dickieanae* p.p. Brullo and Siracusa in Brullo et al. 2004 (Rel. 1–4, 6–12)

Holotypus: Rel.10, Table 22 [15]

Diagnostic species: *Amphidium mougeotii*

Structure and ecology: It is a bryo-pteridophytic community preferring very shaded volcanic rocky habitats at 1800–2100 m of altitude. It is dominated by *Cystopteris dickieana* and *Asplenium septentrionale* and by some mosses, such as *Pohlia cruda*, *Amphidium*

*mougeotii*, and *Tortula subulata*. This new subassociation replaced in altitude the *Pohlio crudae-Cystopteridetum dickieanae* subass. *typicum*, localizing at 1800–2100 m a.s.l. in the deeper rocky ravines of the lava flows.

Distribution: Serra La Nave, Montagnola.

Notes: The cluster analysis divides the *Pohlio crudae–Cystopteridetum dickieanae* community into two distinct groups here reported as two subassociations: subass. *typicum* corresponding to the association described by Brullo and Siracusa [15] for Mt. Etna, and the new subassociation here described as *Pohlio crudae–Cystopteridetum dickieanae* subass. *amphidietosum mougeotii*. This clear division is mainly linked to a different ecology and floristic composition.

3.2.9. *Pohlio crudae–Cystopteridetum dickieanae* Brullo and Siracusa in Brullo et al. 2004 [15] subass. *typicum*

Holotypus: rel. 19, Table 22, Brullo et al. [15].

Diagnostic species: *Cystopteris dickieana*

Structure and ecology: Chomophilous and orophilous vegetation of particularly humid and shaded volcanic rocky habitats. It prefers the mountain range between 1500 and 2200 m a.s.l., settling in particular at the entrance of lava tunnels or in fissures of basaltic rocks. This bryo-pteridophyte community is characterized by some ferns, such as *Cystopteris dickieana* and *Asplenium septentrionale*, and by several mosses, such as *Pohlia cruda*, *P. annotina*, *Amphidium maugeotii*, and *Bartramia pomiformis* [15]. In the subliminar zone of Lamponi cave, it is replaced by the exclusively moss association *Pohlio crudae–Amphidietum mougeotii*.

Distribution: Monte Corruccio, Giarrita, Lamponi cave [15].

3.2.10. *Asplenio septentrionalis–Dryopteridetum filix-mas* Brullo and Siracusa in Brullo et al. 2004 corr. hoc loco

Synonyms: *Asplenio septentrionalis–Dryopteridetum villarii* Brullo and Siracusa in Brullo et al. 2004 [15] *nom. inept*.

Holotypus: rel. 4, Table 24, Brullo et al. [15].

Diagnostic species: *Dryopteris filix-mas*

Structure and ecology: Comophilous and orophilous community of cool and shaded volcanic rocky habitats. This community is dominated by *Dryopteris filix-mas* and *Asplenium septentrionale* and some bryophytes, such as *Pohlia cruda* and *Bartramia pomiformis* [15].

Distribution: Grotta dei Lamponi [15].

Notes: The characteristic species was erroneously identified by the authors as *Dryopteris villari*. Furthermore, the species is indicated as "recorded by mistake" for Sicily [36]. From a careful diagnosis of the samples collected in the Grotta dei Lamponi, the species corresponds to *Dryopteris filix-mas*, a fern quite rare in Sicily. Therefore, we propose the new name *Asplenio septentrionalis–Dryopteridetum filix-mas* corr. hoc loco [*Asplenio septentrionalis–Dryopteridetum villarii* Brullo and Siracusa in Brullo et al. 2004 [15] in Coll. Phytosoc. 28: 481, tb. 24. 2004 (art. 43)].

## 4. Discussions

The high-altitude lava caves preserve some plant communities belonging to the classes *Cladonio digitatae–Lepidozietea reptantis* and *Polypodietea vulgaris*. The first one includes acidophytic, mesophytic, hygrophytic, or hygro-hydrophytic exclusively bryophyte communities occurring on hilly to alpine belts. In the caves of Mt. Etna, this class is represented by communities of the orophilous alliance *Pohlion crudae*. In particular, *Pohlio annotinae–Brachythecietum velutini* and *Pohlietum crudae* subass. *typicum* were found in the liminar zone of the caves, while the *Pohlietum crudae* subass. *timmietosum bavaricae*, *Bartramietum ithyphyllae*, and *Pohlio crudae–Amphidietum mougeotii* also extend into the subliminal zone. The last one is the most widespread association for the cave habitat and for its troglophile character. All these communities are reported only for Mt. Etna; moreover, *Pohlietum crudae timmietosum* and *Pohlio–Amphidietum* are exclusively found within caves. These

bryophyte communities are characterized by some significant taxa with phytogeographical and conservation interest. Several of these species are found in Sicily only on Mt. Etna, where they are mostly distributed at 1700–2100 m of altitude, reaching, in some cases, 2500 m a.s.l. With this regard, we quote the mosses *Brachytheciastrum collinum*, *Grimmia torquata*, *Mielichhoferia mielichhoferiana*, *Amphidium mougeotii*, and *Isopterygiopsis pulchella*. In particular, *Mielichhoferia mielichhoferiana*, considered Near Threatened in Europe [37], was recently assessed as Endangered (EN) in Italy; moreover, many other species found within caves (i.e., *Amphidium mougeotii*, *Pohlia cruda*, *Bartramia ithyphylla*, *Brachytheciastrum collinum*, *Isopterygiopsis pulchella*, *Polytrichastrum alpinum*, *Hymenoloma crispulum*, *Tortula hoppeana*, and *Grimmia torquata* are candidates for the forthcoming Red List of the bryophytes of Sicily (unpublished data). These species, together with others distributed in the higher parts of the volcano, such as *Grimmia fuscolutea*, *G. alpestris*, *G. donniana*, *Mielichhoferia elongata*, *Schistidium flaccidum* at the present, represent the most interesting glacial relicts of the Sicilian bryophyte flora.

The second class (*Polypodietea vulgaris*) groups the chasmo-chomophytic vegetation dominated by bryophytes and pteridophytes, growing in shaded and damp habitats. On Mt. Etna, this class is represented by two alliances: *Bartramio–Polypodion serrati*, grouping thermophilous and chomophytic communities, and *Pohlio crudae–Asplenion septentrionalis*, including the orophilous, chomophytic communities of the supra-Mediterranean and oro-Mediterranean belts. This last alliance includes some communities of the high-altitude volcanic caves of Mt. Etna characterized by some ferns of particular phytogeographical interest, such as *Cystopteris dickieana*, *Asplenium scolopendrium*, and *Polystichum lonchitis*. This last species is very rare and localized in Sicily, where it is exclusively present on Mt. Etna, finding refuge in a small lava tube. This cave, located at around 1900 m of altitude, hosts about 44 mature individuals distributed inside on the low-sloping rock walls and at the base of the cave. Moreover, *Asplenium scolopendrium*, quite widespread in the Sicilian territory on Mt. Etna, is very localized, detected exclusively in the Grotta degli Inglesi, located at 1700 m above sea level, on the northern side of the volcano.

**5. Conclusions**

This study, conducted on the high-altitude caves of Mt. Etna, has highlighted the importance of the cave habitat, as already reported by Guarino et al. [38]. These habitats function as refuge areas for peculiar ptero-bryophyte and bryophyte communities, as well as for some peculiar high-latitude species (arctic montane, boreo-arctic montane, and boreal montane). Many of the species of these communities are very rare and/or endangered, making their populations crucial for the survival of the species in Sicily. Consequently, these communities need to receive attention and protection despite living in a conservative habitat. Conservation management actions should be taken to mitigate human impacts in these ecosystems due, essentially, to the high presence of tourists and excursionists; therefore, the results of this research may be included in the future conservation and management strategies of the Etna Park Authority.

The caves also represent a reservoir of biodiversity, especially for plants with a high microclimate dependence, such as bryophytes and some pteridophytes. Safeguarding cave habitats could improve cave biodiversity and resilience by fostering biological interactions among multiple species and improving ecological functions [4].

**Author Contributions:** Conceptualization, M.P. and S.S.; methodology, M.P. and S.S.; investigation, M.P., G.B., G.M., D.T. and S.S.; data curation, M.P., G.B. and S.S.; data elaboration, S.S. and G.B; writing—original draft preparation, M.P. and S.S.; writing—review and editing, M.P. and S.S. All authors have read and agreed to the published version of the manuscript.

**Funding:** This research was financially supported by the research programme (Line 3 Starting Grant Progetto HAB-VEG cod. 22722132172), funded by the University of Catania and by PIM Mava Foundation.

**Data Availability Statement:** Data is contained within the article.

**Acknowledgments:** The authors thank the Department of Rural and Territorial Development (Catania) for the technical support during the field activities.

**Conflicts of Interest:** The authors declare no conflicts of interest.

## Appendix A  Syntaxonomical Scheme

*Cladonio digitatae-Lepidozietea reptantis* Ježek and Vondrácek 1962
*Diplophylletalia albicantis* Philippi 1963
*Pohlion crudae Privitera* and Puglisi 1996
*Pohlietum crudae Privitera* and Puglisi 1996

- *typicum*
- *timmietosum bavaricae* Privitera and Puglisi 1996

*Pohlio crudae–Amphidietum mougeotii* Privitera and Puglisi 1996
*Bartramietum ithyphyllae* v. Krusenstjerna 1945
*Pohlio annotinae–Brachythecietum velutini* Privitera and Puglisi 1996
*Brachytecio velutini–Asplenietum trichomanis* Brullo and Siracusa in Brullo et al. 2004
*Polypodietea vulgaris* Jurko and Peciar ex Boscaiu, Gergely and Codoreanu in Ratiu et al. 1966 *Anomodonto–Polypodietalia cambrici* O.Bolòs and Vives in O.Bolòs 1957
*Pohlio crudae–Asplenion septentrionalis* Brullo and Siracusa in Brullo et al. 2004
*Pohlio cruda—Cystopteridetum dickieanae* Brullo and Siracusa in Brullo et al. 2004

- *typicum*
- *amphidietosum mougeotii* subass. nova hoc loco
- *polystichetosum lonchitis* subass. nova hoc loco

*Asplenio septentrionalis–Dryopteridetum felix-max* corr. hoc loco

## Appendix B  Floristic List

**Liverworts**
*Cephaloziella divaricata* (Sm.) Schiffn. var. *divaricata*
*Porella cordaeana* (Huebener) Moore
*Reboulia hemisphaerica* (L.) Raddi subsp. *hemisphaerica*
*Scapania compacta* (Roth) Dumort.
*Targionia hypophylla* L.
**Mosses**
*Amphidium mougeotii* (Schimp.) Schimp.
*Bartramia ithyphylla* Brid.
*Bartramia pomiformis* Hedw.
*Brachytheciastrum collinum* (Schleich. ex Müll.Hal.) Ignatov and Hutten
*Brachytheciastrum velutinum* (Hedw.) Ignatov and Huttunen
*Brachythecium salebrosum* (Hoffm. ex F.Weber and D.Mohr) Schimp.
*Ceratodon purpureus* (Hedw.) Brid.
*Coscinodon cribrosus* (Hedw.) Spruce
*Cynodontium bruntonii* (Sm.) Bruch and Schimp.
*Distichium capillaceum* (Hedw.) Bruch and Schimp.
*Distichium inclinatum* (Hedw.) Bruch and Schimp.
*Enthostodon pulchellus* (H. Philib.) Brugués
*Eurhynchium striatum* (Hedw.) Schimp.
*Fissidens bryoides* Hedw.
*Grimmia torquata* Drumm.
*Grimmia trichophylla* Grev.
*Homalothecium sericeum* (Hedw.) Schimp.
*Hymenoloma crispulum* (Hedw.) Ochyra
*Hypnum cupressiforme* Hedw.
*Isopterygiopsis pulchella* (Hedw.) Schimp.

*Isothecium alopecuroides* (Lam. Ex Dubois) Isov.
*Leucodon sciuroides* (Hedw.) Schwägr.
*Microeurhynchium pumilum* (Wilson) Ignatov and Vanderp.
*Mielichhoferia mielichhoferiana* (Funck) Loeske
*Oxyrrhynchium hians* (Hedw.) Loeske
*Philonotis capillaris* Lindb.
*Pogonatum aloides* (Hedw.) P.Beauv.
*Pohlia annotina* (Hedw.) Lindb.
*Pohlia cruda* (Hedw.) Lindb.
*Pohlia elongata* Hedw.
*Pohlia lutescens* (Limpr.) H.Lindb.
*Pohlia nutans* (Hedw.) Lindb.
*Polytrichastrum alpinum* (Hedw.) G.L.Sm.
*Polytrichum commune* Hedw.
*Polytrichum juniperinum* Hedw.
*Polytrichum piliferum* Hedw.
*Ptychostomum capillare* (Hedw.) Holyoak and N. Pedersen
*Ptychostomum imbricatulum* (Müll.Hal.) Holyoak and N.Pedersen
*Racomitrium heterosticum* (Hedw.) Brid.
*Schistidium flaccidum* (De Not.) Ochyra
*Sciuro-hypnum reflexum* (Starke) Ignatov and Huttunen
*Sciuro-hypnum starkei* (Brid.) Ignatov ad Hutten
*Scleropodium touretii* (Brid.) L.F. Koch
*Syntrichia ruralis* (Hedw.) F. Weber and D.Mohr
*Thamnobryum alopecurum* (Hedw.) Gangulee
*Timmia bavarica* Hessl.
*Tortula hoppeana* (Schultz) Ochyra
*Tortula subulata* Hedw.
**Ferns**
*Anogramma leptophylla* (L.) Link
*Asplenium ceterach* L.
*Asplenium onopteris* L.
*Asplenium scolopendrium* L. subsp. *scolopendrium*
*Asplenium septentrionale* (L.) Hoffm. subsp. *septentrionale*
*Asplenium trichomanes* L.
*Cystopteris dickieana* R.Sim
*Dryopteris filix-mas* (L.) Schott
*Dryopteris pallida* (Bory) Maire and Petitm. subsp. *pallida*
*Polystichum lonchitis* (L.) Roth
*Polystichum setiferum* (Forssk.) T.Moore ex Woyn.
**Angiosperms**
*Anthemis aetnensis* Spreng.
*Arabis collina* Ten. subsp. *rosea* (DC.) Minuto
*Cerastium semidecandrum* L.
*Chamaenerion angustifolium* (L.) Scop.
*Crepis leontodontoides* All.
*Festuca circummediterranea* Patzke
*Galium aetnicum* Biv.
*Hypochaeris laevigata* (L.) Ces., Pass. and Gibelli
*Luzula forsteri (Sm.)* DC.
*Petrorhagia saxifraga* (L.) Link subsp. *gasparrinii* (Guss.) Pignatti ex Greuter and Burdet
*Petrosedum tenuifolium* (Sm.) Grulich
*Rubus idaeus* L. subsp. *idaeus*
*Rumex scutatus* L. subsp. *aetnensis* (C.Presl) Cif. and Giacom.

*Sedum dasyphyllum* L.
*Sedum hispanicum* L.
*Senecio aethnensis* Jan ex DC.
*Silene italica* (L.) Pers. subsp. *sicula* (Ucria) Jeanm.
*Tanacetum vulgare* L. subsp. *siculum* (Guss.) Raimondo and Spadaro
*Umbilicus rupestris* (Salisb.) Dandy

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
