# Peer review of "The High-Altitude Volcanic Caves of Mount Etna (Eastern Sicily): A Relevant Refuge for Some Ptero-Bryophyte Communities"

_land, doi:10.3390/land13070967_

Round 1

Reviewer 1 Report

Comments and Suggestions for Authors

Article dealing with the phytosociology of bryophytic and pterophytic vegetation in the volcanic caves of Etna. Interesting work of relevance for dealing with fragile habitats of conservation interest. I congratulate the authors for this work and encourage them to continue with this line of research. In order to improve the study, I suggest including the authorship of the taxa, since they have not been included. In addition, a joint section on Results and Discussion has been written, in my opinion they should be separated, since the results are very clear, but not so much the discussion.

The research enters into conservation fragile areas. The authors present the type of habitats following the phytosociological methodology. The work is not excessively original, but the diagnosis of plant associations to discern the habitat to which they belong according to the EU Habitats Directive is interesting. It provides a better knowledge of the plant associations, with the possibility to study the biodiversity of volcanic caves. The methodology from my point of view is correct, it is a very well known methodology in the botanical world, it could be considered that the authors should consider how to establish the conservation status of the habitats. The conclusions are correct, although they may be subject to modification when the authors break down results and discussion. The references are appropriate, but should be expanded when the discussion section is drafted. In this new discussion section I suggest that the authors make a comparative analysis with other areas similar to Etna. Tables, figures, photographs are correct.

Author Response

Dear REV

all the requested suggestions have been integrated into the text

tks for all

SS

Reviewer 2 Report

Comments and Suggestions for Authors

The manuscript provides a detailed description of plant communities of the volcanic caves of Etna Mountain, Sicily. This highly specialized habitat has its unique role in biodiversity conservation, but also in understanding of vegetation processes; thus, it could bring novel and challenging information for many readers.

The content of the manuscript  is of high quality, although slightly unbalanced toward descriptive side. It followed the specific methodology for this type of study. The number of samples and the statistical analysis is adequate. 

Minor issues were detected for which detailed suggestions and comments are provided in the attached file. The most important issues to address are: 

- if the ordination was meant to validate the classification, please give details about the ordination method applied; consider the inclusion of an ordination diagram on appendix; alternatively, you may refer to the cophenetic correlation or silhouette index

-  consider to add a synoptic table to facilitate comparison of species composition among plant communities; this table could serve as classification validation; it may appear on appendix.

- split the present Conclusions into Discussions and Conclusions; most of the information here is either too descriptive, detailed or untouched previously

- indicate clearly which are the novelties in this study comparing with previous ones

Author Response

(The authors gave the same response as above.)

Reviewer 3 Report

Comments and Suggestions for Authors

This paper aims to identify ptero-bryophyte communities in the volcanic caves on Mount Etna. The data include phytosociological records collected both in the field and from the literature.

This study holds significant phytosociological importance as it describes the associations of understudied plant groups, such as bryophytes. It can greatly contribute to habitat descriptions for future investigations and enhance our understanding of the distribution of bryophytes and pteridophytes in specific conditions, such as volcanic caves.

I recommend accepting this paper after minor revisions.

Specific comments: 

Lines 61-64. Please provide references (You said "some researchers.....", add references after that statement).

Lines 76-77. Provide some information about the sampling methodology. How did you collect the relevés, what is the plot size, how did you place the sampling plots, etc.

Author Response

(The authors gave the same response as above.)
